# The Role of Smart Technology in Sustainable Agriculture: A Case Study of Wangree Plant Factory

**Salinee Santiteerakul** [1,2], **Apichat Sopadang** [1,2,*], **Korrakot Yaibuathet Tippayawong** [1,2] **and Krisana Tamvimol** [3]

1    Center of Excellence in Logistics and Supply Chain Management, Chiang Mai University,
Chiang Mai 50200, Thailand; salinee@eng.cmu.ac.th (S.S.); korrakot@eng.cmu.ac.th (K.Y.T.)

2    Department of Industrial Engineering, Faculty of Engineering, Chiang Mai University,
Chiang Mai 50200, Thailand

3    Wangree Health Factory Co., Ltd., Nayok 26000, Thailand; krisana@oi.co.th

*    Correspondence: apichat.s@cmu.ac.th

**Abstract:** Sustainable development is of growing importance to the agriculture sector because the current lacking utilization of resources and energy usage, together with the pollution generated from toxic chemicals, cannot continue at present rates. Sustainability in agriculture can be achieved through using less (or no) poisonous chemicals, saving natural resources, and reducing greenhouse gas emissions. Technology applications could help farmers to use proper data in decision-making, which leads to low-input agriculture. This work focuses on the role of smart technology implementation in sustainable agriculture. The effects of smart technology implementation are analyzed by using a case study approach. The results show that the plant factory using intelligence technology enhances sustainability performance by increasing production productivity, product quality, crop per year, resource use efficiency, and food safety, as well as improving employees' quality of life.

**Keywords:** smart technologies; sustainable agriculture; plant factory

---

## 1. Introduction

In the era of the fourth industrial revolution (Industry 4.0), new technologies, such as artificial intelligence, automation systems, and cloud computing have been developed to combine the digital and physical world together. One of the Industry 4.0 benefits is to provide a crossing workspace among machines, infrastructures, and digital platforms. [1]. The methods of Industry 4.0 have been investigated in the area of agriculture. Various innovations, like sensor technology, machine learning, wireless communication, positioning systems, and data visualization tools, have been adopted to create value and increase productivity in the farming sector [2].

Due to the environmental stress on water scarcities, insufficient land use, soil depletion, and greenhouse gas emissions, the demands on sustainable agriculture are rapidly increasing. The sustainable development (SD) concept was developed regarding a reorganization of a threatened future. Many regions face risks of inevitable damage to the human environment. Environmental stress has been seen as the result of the increasing growth of the population, technology growth and development, and the rising living standards among the affluent. Given the importance of agriculture as the crucial provider of food, the sustainable development of this sector is important. Thus, sustainable agriculture needs an innovative system that protects and enhances the natural resource base while increasing productivity.

There are essential research works that deal with aspects of the technology implementation for sustainable agriculture. The authors of [3] Reviewed relevant research works on intelligent agricultural information handling methods. The technological applications related to agricultural aspects were classified into three categories, namely data sources and collection, machine learning (ML) methodologies for agricultural data, and intelligent knowledge acquisition. These applications lead to the development of innovation for precision and optimal farming. The authors of [4] studied artificial devices of closed plant production systems and found that smart plant production systems produce high-quality plants and transplants with minimum use of resources and carbon dioxide emissions as well as environmental pollutants.

This work aims to study the effects of smart technology implementation on sustainability performance by using a case study. The case study company, namely the Wangree Health Factory Company (located in Nakhon Nayok, Thailand), deploys plant factory with artificial lighting technology to cultivate fresh organic vegetables and fruits. This work develops the analysis framework to analyze how deploying plant factories will impact the sustainability of agriculture. This article is organized as follows: first, the literature review provides a background on smart farming and sustainable agriculture. Then, the material and methods section provides a research framework and a case study analysis, and is followed by the results section. Finally, the roles of smart technology are presented and discussed.

## 2. Literature Review

### 2.1. Smart Farming

Smart farming is a management concept using modern technology to increase the quantity and quality of agricultural products. Smart farming involves an integration of information and communication technologies into machines, sensors, actuators, and network equipment for use in agricultural production systems [5]. There are several technologies related to smart farming, including sensors, robotics, the Internet of Things (IoT), mapping, decision-making, and statistical processes [5,6].

Ray [6] proposed the detailed framework of IoT-based agriculture. This framework comprises six layers. First, the physical layer contains various types of devices, such as sensors and microcontrollers, to collect, exchange, and process data to other devices. Second, the network layer comprises the Internet and relevant communication technologies. Third, the IoT-based middleware layer performs various tasks, such as device management, interoperation, context awareness, platform portability, and security-related tasks. Fourth, the service layer provides cloud storage and Software-as-a-Service (SaaS). Fifth, the analytics layer performs big data processing to predictive and multi culture analysis. Sixth, the user experience layer facilitates the farmer's communication using social network activities to share and disseminate agricultural knowledge.

Plant factories are one of the smart farming applications. A plant factory is a closed-growing system that enables a farmer to achieve constant and regular production of vegetables throughout the year. There are three types of plant factories: (1) Plant factory with sunlight; (2) Plant factory with sunlight and supplemental light; and (3) Plant factory with artificial lighting. The plant factory with artificial lighting replaces sunlight with artificial intelligence sources of light. This plant factory creates a more consistent light environment for the plants. Ray [6] explained that there are three units for managing plant factory systems: (1) Farm Gate Way (FGW), (2) A data collection/storage/distribution platform, and (3) an application module. A FGW unit is a data collection and control device. It collects growing conditions data in the plant factory (such as temperature and nutrient content) and the crop production equipment data (such as nutrient solution pumps and heat pumps). A cloud-based data collection, storage, and distribution platform is used to provide communication between the data center and the plant factory for the control of growing conditions and equipment. It also helps in the servicing of command and control information management. The application unit is composed of physical devices to perform three operations: sensory data reception, command delivery to the FGW, and response reception from the FGW.

This work employed the integrated IoT-based agriculture and plant factory unit framework (see Figure 1) to analyze the technology implementation of a case study company.

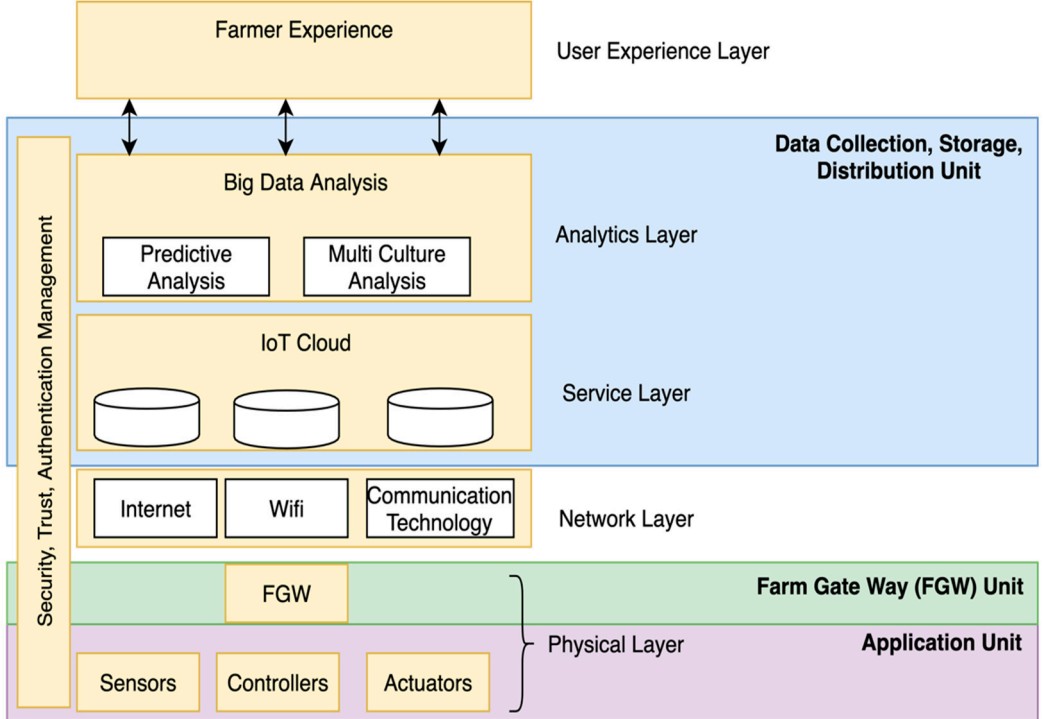

**Figure 1.** Internet of Things (IoT)-based agriculture and plant factory units framework (modified from [6]).

### 2.2. Sustainable Agriculture and Sustainability Measurement

Over the past decade, societies have changed rapidly as a result of technology advancement, rapid urbanization, and innovations in production systems. At the same time, enormous changes have occurred, such as climate change, an increasing number of crises and natural disasters, increasing of population dynamics, and economic growth. Crop production is damaged by floods, climate change, drought, and insufficient land resources. The world's population is expected to increase by more than 30% by 2050 (from 7 billion in 2011). Meanwhile the global crop production is expected to grow by more than 90% from higher yields. The world's agricultural production has increased more than three times between 1960 and 2015 [7]. This caused the use of resource conservation technologies and productivity enhancement for agricultural purposes. These trends threaten the sustainability of agricultural systems and undermine the global capacity to meet its needs.

A useful definition of sustainability is that of the World Commission on Environment and Development (1987), which states: "A sustainable economy is one which can meet the needs of the present without compromising the ability of future generations to meet their own needs" [8]. Sustainability as an attribute of agriculture has been gaining recognition globally since the 1970s [9]. The term "sustainable agriculture" refers to an agricultural system that will continue to be productive in the future.

There are many suggested measures of sustainability. Most of the literature uses the triple bottom line (TBL) developed by Elkington [10]. This concept divides sustainability into three categories, including economic, environment and society. The sustainability indicators are developed based on the TBL concept by defining a set of indicators to indicate the performance of each category [11–16].

At the farm level, numerous sustainability assessment methods have been developed to assess the sustainability performance of agricultural systems. Some examples of sustainability assessment methods from the scientific literature are the sustainability assessment of food and agriculture systems

(SAFA) [13], sustainability monitoring and assessment routine (SMART) [14], response-inducing sustainability evaluation (RISE) [17], and multi-criteria assessment (MCA) [18].

The indicators of measuring the sustainability of the agriculture system are divided into three main categories. The first category is technical data, which require specific quantitative data, such as energy consumption, water use, soil pH, and farm revenue. The second category is the performance rating indicator, which requires an approach to convert sustainability performance to rating scores. For example, to evaluate working condition performance, the working environment is assessed in terms of a 5-scale rating from worse (score = 1) to excellent (score = 5). The third category is a descriptive indicator, which relates to the driver, pressure, state, impact, or response of sustainability. Quantitative and qualitative descriptive indicators describe the factual situation but do not assess whether this is good or bad—they are, in practical terms, a statement of a fact.

## 3. Material and Methods

### 3.1. Research Framework

A framework to analyze the roles of smart technology in sustainable agriculture consists of four steps (see Figure 2).

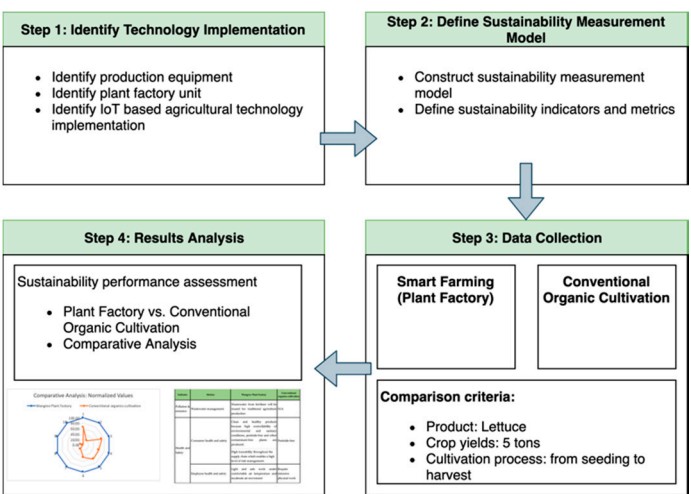

**Figure 2.** Research Framework.

Step 1: Identify technology implementation of the case study by using the IoT-based agriculture and plant factory unit framework [6].

Step 2: Define a sustainability measurement model by defining sustainability indicators and metrics and constructing the measurement model.

Step 3: Planting data was collected by using the case study. The data on cultivation in a plant factory were collected from Wangree Health Factory Company. The data on conventional organic cultivation were collected from Wangree Organic Farm. The relevant data were collected based on the assumption of equal production outputs, which is one-crop cultivation of 5 tons of product weight. This data collection includes all activities from seeding to harvest process.

Step 4: Assess the sustainability performance of plant factory cultivation compared to conventional organic cultivation.

### 3.2. Case Study Overview

Wangree Health Factory Company, located in Nakhon Nayok, Thailand, was founded in 2016 with the corporate vision of using modern digital technology to provide fresh organic vegetables and fruits to the Thai market. A plant factory with artificial intelligence light is an indoor farming system

connected with a smart control system. The structures of the plant factory separate the plants from the external environment so that the plants are protected from uncertain conditions. These systems permit high-quality and high-yield production year-round under a controlled environment (e.g., light, temperature, humidity, the concentration of carbon dioxide, and culture solution). The IoT-based technologies allow farmers to plan their production by using mobile devices for monitoring and controlling their farming systems.

Wangree Health Factory Company combines a vertical farming system with IoT technologies (see Figures 3 and 4). The growth process is fully automated for watering, lighting, nutrient adding, and temperature controlling. The 173.85 m$^2$ × 6 m high plant factory produces approximately 50,552 heads of lettuce per month. The main production equipment is composed of electricity supply, an air conditioning system, nutrient solution supply, lighting, $CO_2$ supply, smart devices, and communication devices.

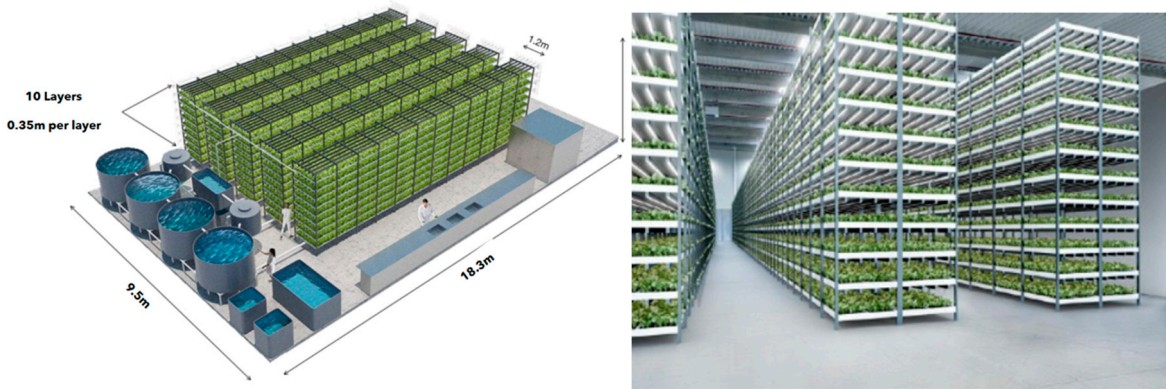

**Figure 3.** Plant factory, a vertical farming system.

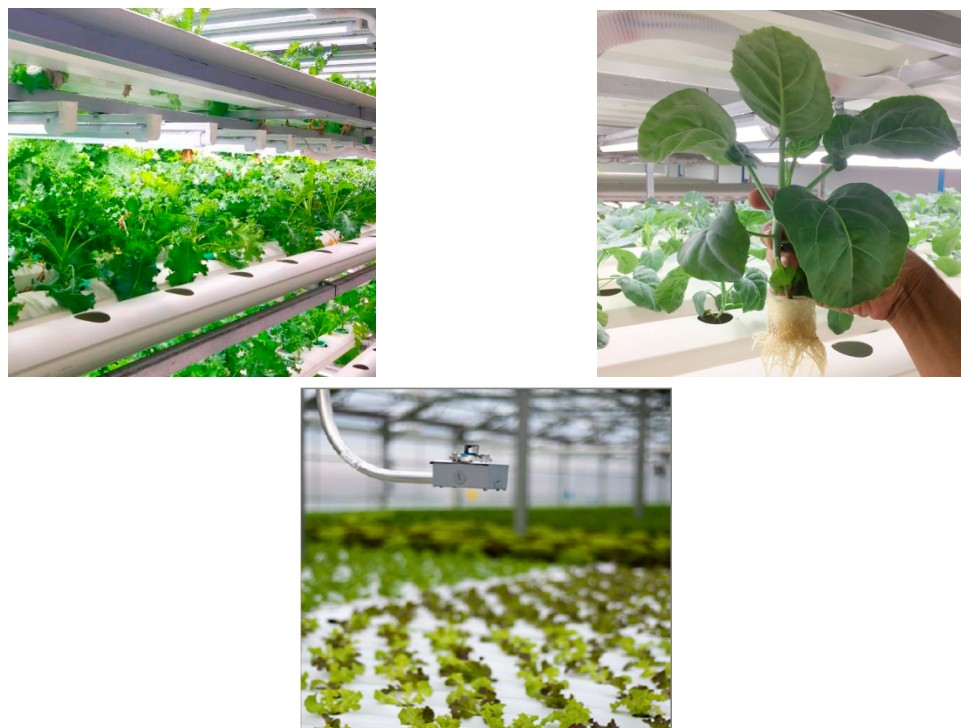

**Figure 4.** Seedlings grow with light at Wangree Health Factory Company.

Wangree Plant Factory employs IoT-based technology to produce and manage the agriculture production system. The workflow of the system consists of three steps (see Figure 5): (1) Detecting and collecting data from sensor devices; (2) Analysis and control based on a specific AI algorithm; and (3) Data visualization for statistical reporting to help farm owners make decisions.

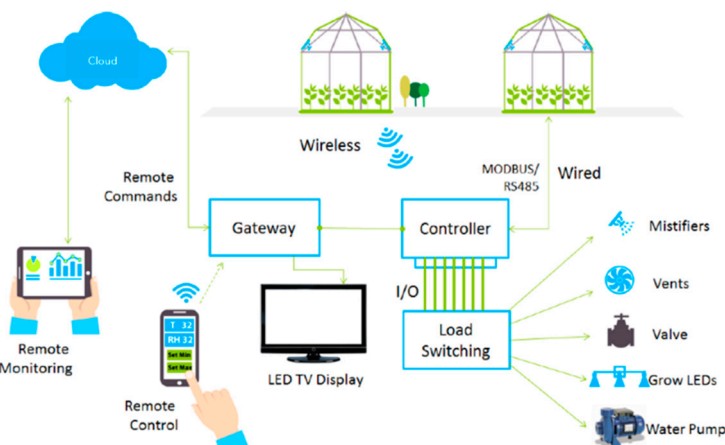

**Figure 5.** Smart technology implementation at Wangree Health Factory Company.

The lists of equipment for one plant factory are shown in Table 1. The hardware equipment comprises seven categories, i.e., electricity supply, air conditioning, nutrient solution supply, lighting, $CO_2$ supply, sensors for environmental control, and communication and management equipment.

**Table 1.** Equipment and environmental sensors typically installed in a plant factory.

| Category | Equipment and Environmental Sensors |
|---|---|
| Electricity supply | Power distribution box and backup power system<br>Power Consumption AC 100 V/240 V 100–150 W (Power Saving Mode),<br>300–350 W (Full Power Mode) |
| Air conditioning | Inner units of air conditioners 40,000 BTU *<br>Air circulation fans<br>Air cleaners with filters |
| Nutrient solution supply | Water and fertilizer system<br>Cultivation controller system<br>Pest protection system<br>Plumbing for clean water supply<br>UV water purifier |
| Lighting | Light source with reflectors<br>LED lumen 5 cm—11,000 lum and 26 cm—9000 lum |
| $CO_2$ supply | Control unit with distribution tubes |
| Sensors for environmental control | Smart light control<br>Smart air sensors<br>Smart water feeding<br>Temperature and humid sensors<br>$CO_2$ sensors |
| Communication and Management | Wireless communication<br>Plant dashboard visualization |

* Note: British Thermal Unit (BTU) is a unit of heat. It is defined as the amount of heat required to raise the temperature of one pound of water by one degree Fahrenheit. One BTU is about 1055 Jules.

The software system (see Figure 6) comprises three main functions: (1) a cloud data center system with an intelligence processing system (Big Data and AI) is responsible for storing databases for

automatic processing and backup of essential data. (2) A back-end information management system, which is a web-based system, is responsible for managing information in terms of data security, right of data accessing, and data editing. (3) A mobile application system for both iOS and Android platforms is an application module used for managing and controlling operational production. It deploys remote controlling technology and allows users to monitor and manage their operation in real-time from a distance via smartphone and tablet devices.

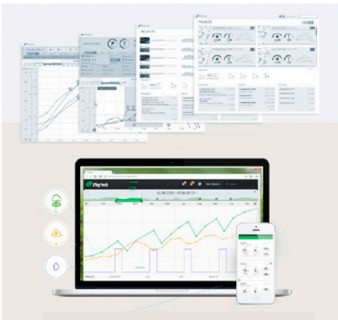 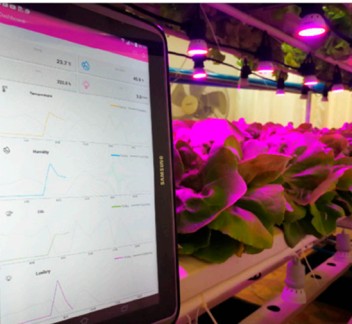

**Figure 6.** Software system of Plant Factory at the Wangree Health Factory Company.

Following the framework developed by Ray [6], the IoT-based technology implementation is shown in Figure 7.

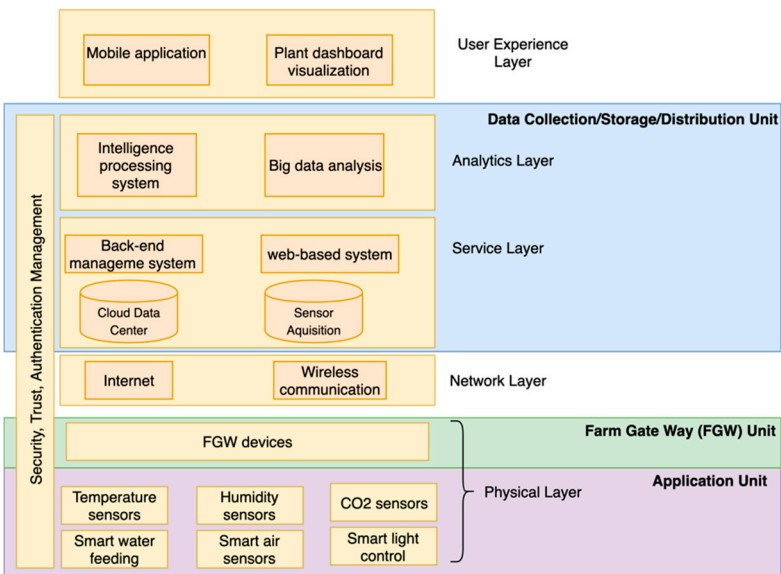

**Figure 7.** Wangree Health Factory Company technology implementation.

### 3.3. Sustainability Measurement Model Definition

This work adopts the sustainability performance measurement model proposed by [19] (see Figure 8). The objective of the proposed model is to measure the sustainability performance of sustainable farming. This model is composed of five levels. Level 0 is the top level and identifies the goal, which is sustainable agriculture.

Level 1 is a dimension level dividing sustainability into economic, environmental, and social categories. Level 2 is a sub-dimension level. There are eight sub-dimensions regarding three principal dimensions. For example, raw material, natural resources, and energy are sub-dimensions of the environmental perspective. Level 3 is an indicator level that provides a performance indicator to indicate the performance of each sub-dimension. For example, we could evaluate the impact of natural

resources, from resource consumption to pollution and emission. Level 4, the last level, is a metric level that provides a formulation to measure sustainability performance. For example, resource consumption performance is indicated by the quantity of water consumption, land use, and soil quality.

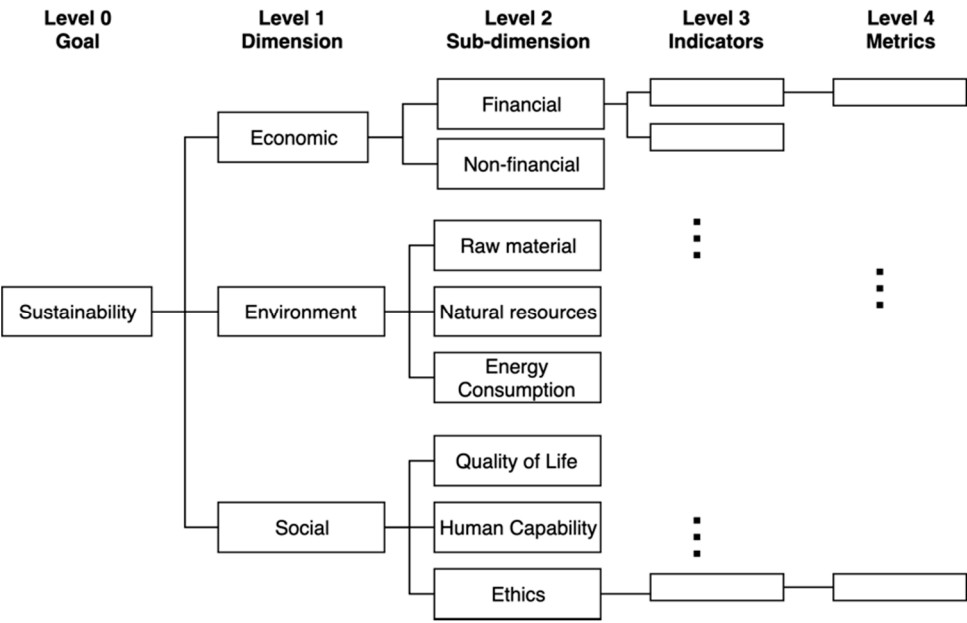

**Figure 8.** Sustainability performance measurement framework.

To implement the sustainable agriculture performance measurement model, [20] suggested that decision-maker(s) should use level 0 to level 2 as a framework to select specific indicators (level 3) and metrics (level 4) that are appropriate for their application. Therefore, this work will construct the performance indicators and metrics that relate to smart technology implementation in agriculture.

This work selected seven sub-dimension (level 2) and the details are as follows:

- Financial criteria relate to increasing revenue, profit, market share, reducing the cost of operation, and unit cost.
- Non-financial criteria relate to improving product quality, crop per year, and reducing harvest time.
- Raw material criteria relate to conserving and enhancing the raw material by efficient use through the reduce, reuse, and recycle concepts; reducing hazardous material usage; and reducing defects and waste generation.
- Natural resource (water, soils, and land use) criteria relate to conserving and enhancing the natural resource base by efficiently using and reducing environmental emission.
- Quality of life criteria relate to assessing and reducing the potential health impacts of new technologies as well as increasing the well-being of stakeholders.
- Human capability criteria relate to encouraging education to improve human skills and knowledge performance.
- Ethics criteria relate to respecting local and international laws on business and human rights and supporting ethical operating practice issues.

The details of sustainability indicators and metrics are shown in Table 2.

**Table 2.** Sustainability indicators and metrics.

| Dimension | Sub-Dimension | Indicator | Metrics |
|---|---|---|---|
| Economic (EC) | Financial | Cost | Infrastructure cost (EC-1) Fertilizer cost (EC-2) Unit cost (EC-3) |
| | Non-Financial | Productivity | Harvest time (EC-4) Weight per unit (EC-5) Product quality grade (EC-6) Crop per year (EC-7) Lifetime of cultivation system (EC-8) |
| Environmental (EN) | Raw material | Defects | Percentage of defects (EN-1) |
| | Natural resource | Resource consumption | Water use (EN-2) Land use (EN-3) |
| | | Pollution and emission | Wastewater management (EN-4) |
| Social (SO) | Quality of life | Health and Safety | Consumer health and safety (SO-1) Employee health and safety (SO-2) |
| | Human capability | Knowledge sharing | Social support by sharing knowledge to the local community (SO-3) |
| | Ethics | Fair operation practices | Fair salary (SO-4) |

## 4. Results

The results of the sustainability performance of plant factories and conventional organics cultivation are shown in Table 3.

**Table 3.** Results of comparing sustainability Performance.

| Indicator | Metrics | Wangree Plant Factory | Conventional Organics Cultivation |
|---|---|---|---|
| Cost | Infrastructure cost (million baht) | 5–6 | 10–12 |
| | Fertilizer cost (baht) | 10,000 | 50,000 |
| | Unit cost (baht) | 4.25 | 6.00 |
| Productivity | Harvest time (days) | 21–30 | 45–50 |
| | Weight per unit (g) | 100–175 | 75–100 |
| | Product quality grade | Medical grade (post organic) | Good agricultural practice (GAP) or Organic |
| | Crop per year (crops) | 12–15 | 4–8 |
| | Lifetime of cultivation system (years) | 15–30 | 3–5 |
| | Labors (man) | 2 | 10–20 |
| Defects | Percentage of defects (%) | 0.5–1% | 30–50% |
| Resource consumption | Cultivation water use for (liter per month) | 30,000 | 3,000,000 |
| | Land use (m$^2$) | 160 | 1600 |
| Pollution and emission | Wastewater management | Wastewater from fertilizer will be reused for traditional agriculture production. | N/A |
| Health and Safety | Consumer health and safety | Clean and healthy products: because of the high controllability of the environmental and sanitary conditions, pesticide-free and other contaminant-free plants are produced. High traceability throughout the supply chain, which enables a high level of risk management. | Pesticide-free |
| | Employee health and safety | Light and safe work under comfortable air temperature and moderate air movement. | Requires intensive physical work |
| Knowledge sharing | Social support by sharing knowledge to the local community | Wangree Plant Factory provides knowledge sharing to the local community, such as researchers in academic institutes, governance institutes, and private companies. | N/A |
| Fair operation practices | Fair salary | Due to the requirements of highly skilled persons and less labor needed (only 2–3 persons compared with 10–15 persons for traditional organic production), the Wangree Plant Factory could pay a higher salary to its employees. | N/A |

Wangree Plant Factory has actualized most of the potential benefits in the economic and environmental aspects. Compared with traditional organic cultivation, the Wangree Plant Factory can greatly reduce the consumption of resources than conventional cultivation. The percentage of resources saved and the improvement in produce quality and yield in the plant factory are as follows:

**Economic Perspective:**

- A 50% reduction of infrastructure cost and a 3–5 times reduction of lifetime cultivation system.
- A 80% reduction of fertilization cost because of lower fertilizer consumption achieved through recycling with little drainage of circulating nutrient solution.
- A 30% reduction of unit cost because of lower resource consumption and higher productivity.
- A 33–75% increase of product weight per unit.
- An increase by 1.8–2 times of amount of a crop per year.
- Better product quality grade.

**Environmental Perspective:**

- Both the plant factory and organic cultivation are free-pesticide applications. The plant factory keeps the cultivated area clean and free from pest insects.
- A 99% reduction in water consumption.
- A 99% reduction in land use compared to conventional agriculture due to a higher productivity per production area.
- A 30–50% reduction in plant defects.

**Social Perspective:**

- Increasing demands for fresh food, nutritious food, and functional food for health care and higher quality of life because of high controllability of plant environment. Controlled aerial environmental factors include photosynthetic photon flux density (PPFD), air temperature, $CO_2$ concentration, light quality, and flow rate of the nutrient solution.
- Light and safe work under comfortable air temperature and moderate air movement.

## 5. Discussion

The closed environment of a plant factory isolates crop production from an exterior environment. Consequently, the plant growth factors, such as water, light, carbon dioxide concentration, and nutrients are controllable. A plant factory control system can measure the environmental conditions, such as temperature, humidity, and light intensity, and respond by adjusting the plant growth factors via controllers. Measurement and responsiveness capabilities depend on the sensors' data collection and storage and on the processing ability of cloud computing with AI and Big Data analysis. Moreover, the effective management system and data visualization encourage farmers to make good decisions to operate their production.

This work studies how technology affects sustainable agriculture. The sustainability performance is composed of three dimensions: economic, environment, and society. The effects from technology implementation in each dimension are discussed here.

**Economic Dimension**

- Increasing product quality: precision measurement and a suitable plant growth factor adjustment lead to better product quality. It can be seen that the products from the plant factory are higher in weight per unit, and have a better quality grade and a lower percentage of defects.
- Increasing productivity: the production productivity of the plant factory is higher than the conventional cultivation due to the cost reduction from resource use (water and fertilization), the increase of product weight per unit, and the amount of labor reduction.

- Increasing crop per year: the plant factory has an artificial intelligence system completely closed off the outside environment. It replaces sunlight with controllable lighting sources and controls other plant growth factors, such as humidity, carbon dioxide concentration, temperature, and nutrients by using an artificial intelligence system. Consequently, a plant factory can achieve a year-round production environment. The plant factory produces around twice as much crop per year compared to conventional agriculture and reduces harvest time by around 50 percent compared to traditional cultivation.

### Environmental Dimension

- Increasing resource use efficiency: the plant factory enhances crop irrigation water productivity due to a water control system that reduces drained water in the growing area and recycles water vapor into liquid water. The vertical farming of the plant factory increases land use efficiency. It provides a 99% reduction in land use.

### Social Dimension

- Increasing food safety: the plant factory gives priority to keeping the growing area free from pests and pesticides. These hygiene conditions create a ready-to-eat product after harvesting. Moreover, the information technology in the plant factory allows customers and stakeholders in the supply chain to trace operational data from producers.
- Increasing employees' quality of life: the controllable working environment in the plant factory is much more desirable than field cultivation, which involves the uncertainty of heat and weather. Further, to work with automatic and high technology systems, the plant factory requires highly skilled workers. It encourages employees to improve their skills and knowledge.

It is clear that the plant factory with artificial system enhances the sustainability performance in all economic, environmental, and social perspectives. The isolated environment of the plant factory provides numerous advantages for improving productivity, improving product quality, increasing resource use efficiency, and increasing food safety. However, plant factories tend to require a greater energy input. Due to the fact that free energy from sunlight has been rejected from this growing system, the plant factory needs new energy to provide a light source to the system. It This leads to much higher costs of lighting in the plant factory. The electricity cost represents nearly one-fourth of the production costs [21] This issue can be solved and further investigated in future research.

## 6. Conclusions

Technological development and digitalization shape feasible boundaries to increase resource use efficiency. Smart agriculture reduces the negative environmental impacts of farming, increases resilience and soil health, and decreases costs for farmers. The number and types of challenges associated with smart farming expand across various agricultural production systems, and infrastructural limitations apply when it comes to IoT implementation. The plant factory is one of the solutions to solve the problems regarding foods, resources, and the environment. Methodologies have been developed by which the yield and quality of foods are improved, with less consumption of resources and less environmental degradation than the current plant production system. The potential benefits of the plant factory are enhanced economic and environmental sustainability.

**Author Contributions:** A.S. supervised the work and worked on the selection and application methodology on the real case study. S.S. contributed the whole structure of this paper, worked on the literature review and the development of research framework, and summarized the findings and the conclusion. K.Y.T. contributed on the project coordinating and worked on reviewing and editing manuscript. K.T. contributed the case study data. All authors have read and agreed to the published version of the manuscript.

**Funding:** This research and the APC were funded by the project "SME 4.0—Industry 4.0 for SMEs" (funded by the European Union's Horizon 2020 R&I program under the Marie Skłodowska-Curie grant agreement No. 734713).

**Acknowledgments:** This research work was partially supported by Center of Excellence in Logistics and Supply Chain Management, Chiang Mai University and Wangree Health Factory Co., Ltd., Nakhon Nayok, Thailand.

**Conflicts of Interest:** The authors declare no conflict of interest.

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
