# Peer review of "The Role of Smart Technology in Sustainable Agriculture: A Case Study of Wangree Plant Factory"

_sustainability, doi:10.3390/su12114640_

Round 1

Reviewer 1 Report

Unfortunately, the manuscript does not fulfill the basic requirements for publication.

In its current form, the manuscript requires a major English revision. A professional help by a native EN speaker could help, but often verbs and adjectives are misused, and it might require more careful work from the authors.

Moreover, another major flaw that makes the publication of this ms impossible in this form, is that the M&M and Results sections do not follow the author guidelines. Particularly, M&M includes extensive information, some of which would be useful in the introduction, but none of them refer to the current study. Instead, the results section presents the case study (L239), while this should have been done in M&M. In this section, only the results of this work had to be mentioned.

Finally, Figs.1, 2, and 3 are taken from different articles and this should have been specified more clearly to avoid copyright violations. Other figures, such as fig.9 are totally superfluous. Another fatal issue is that there is no discussion in this article, although this is a mandatory section according to the author guidelines. Instead, there are conclusions (facultative), but those cannot be really considered reliable because of the flaws above. References are not written in a consistent form. I provided minor suggestions and changes in the pdf documents.

Reviewer 2 Report

This study is to understand the effects of smart technology implementation based on a case study. The overall all comments on this manuscript is to reorganize the structure. It is not very clear that if this work is a review paper, or a research paper. If it is a research paper, the information from the Methods section should be moved to introduction. And the first part of the Results section should be used as method. Also, the Discussion section is missed. If it is a review paper, then a method on how literature was collected should be given.

Reviewer 3 Report

This article is a case study that discussed the role of smart technology for agricultural sustainability. However there are several issues regarding how the manuscript has been presented. The abstract needs to be rewritten. It should include some information from discussion and overall conclusion from the study.

Enough background for the case study is not provided. The objective of this study is not clear. What are the knowledge gaps? There is a lot of information in the paper but lacks organization and structure.

Is the discussion session missing or merged with Results? This article needs major formatting. This will increase readership.

Round 2

Reviewer 2 Report

The manuscript has been greatly improved. The format and language needs to be carefully checked before publication.